# Evaluation of tarsal injuries in C57BL/6J male mice

**Brenda L. Kick**[1¤]\*, Laura Anderson[2], Rosalinda Doty[1], Christine Wooley[2], Meaghan Dyer[2], Torrian Green[2], Veronica Knickerbocker[2], Zoe Brown[2], Samantha Loeber[3], Janine Wotton[2], Bonnie Lyons[1], Linda Waterman[1], Zoë Bichler[2]

1 Comparative Medicine and Quality, The Jackson Laboratory, Bar Harbor, Maine, United States of America,
2 Center for Biometric Analysis, The Jackson Laboratory, Bar Harbor, Maine, United States of America,
3 Department of Surgical Sciences, School of Veterinary Medicine, University of Wisconsin, Madison, Wisconsin, United States of America

¤ Current address: Research Animal Resources, University of Minnesota, Minneapolis, Minnesota, United States of America
* bkick86@gmail.com

**Data Availability Statement:** All relevant data are within the paper and its Supporting information files.

**Funding:** The authors received no specific funding for this work.

## Abstract

Tarsal joint abnormalities have been observed in aged male mice on a C57BL background. This joint disease consists of calcaneal displacement, inflammation, and proliferation of cartilage and connective tissue, that can progress to ankylosis of the joint. While tarsal pathology has been described previously in C57BL/6N substrains, as well as in STR/ort and B10. BR strain, no current literature describes this disease occurring in C57BL/6J mice. More importantly the behavioral features that may result from such a change to the joint have yet to be evaluated. This condition was observed in older male mice of the C57BL/6J lineage, around the age of 20 weeks or older, at a frequency of 1% of the population. To assess potential phenotypic sequela, this study sought to evaluate body weight, frailty assessment, home cage wheel running, dynamic weight bearing, and mechanical allodynia with and without the presence of pain relief with morphine. Overall mice with tarsal injuries had significantly higher frailty scores (p< 0.05) and weighed less (p<0.01) compared to unaffected mice. Affected mice had greater overall touch sensitivity (p<0.05) and they placed more weight on their forelimbs (p<0.01) compared to their hind limbs. Lastly, when housed with a running wheel, affected mice ran for a shorter length of time (p<0.01) but tended to run a greater distance within the time they did run (p<0.01) compared to unaffected mice. When tested just after being given morphine, the affected mice performed more similarly to unaffected mice, suggesting there is a pain sensation to this disease process. This highlights the importance of further characterizing inbred mouse mutations, as they may impact research programs or specific study goals.

## Introduction

Recently tarsal injuries have been noted in the C57BL/6N background strains from the International Mouse Phenotyping Consortium (IMPC) (www.mousephenotype.org), which is developing genetically altered mice to study mammalian gene function [1]. At the different

**Competing interests:** The authors have declared that no competing interests exist.

**Abbreviations:** ANKENT, Ankylosing Enthesiopathy; DWB, Dynamic Weight Bearing; MicroCT, Micro Computed Tomography.

phenotyping centers, abnormal tarsal joints were noted at various rates in group housed male mice, ranging from 1.7% up to 12%, and beginning from 4 weeks of age to 78 weeks. Similar to the spontaneous ankle deformities in STR/ort mice and ankylosing enthesopathy in B10.BR strain, this condition was not found in females of the same strain and age, or in singly housed males or males in breeding groups [2, 3]. The exact reasons for why there is such a wide range of ages and total percentage of the population affected between the IMPC remains to be known. Some hypotheses may be microbiome differences between the institutions, housing and animal care regime differences, or background differences (some of the mice were from Charles River Laboratories lineage, some from Taconic lineage, and some of the mice had Jackson Laboratory lineage). Due to the reported finding of abnormal tarsal joints in the C57BL/6N colony within the IMPC's Knockout Mouse Program at The Jackson Laboratory, this facility further observed their aged C57BL/6J colony. Abnormal tarsal joints were noted in older male mice of the C57BL/6J lineage, around the age of 20 weeks or older, at a frequency of 1% of the population. Like previously reported strains, aged matched female mice were not noted to have any tarsal or other joint abnormalities.

Currently this is the first known report of calcaneal dislocation occurring specifically in the C57BL/6J strain. This study sought to assess the potential phenotypic sequela resulting from this disease. With the C57BL/6J mouse being the most widely used background strain in research laboratories, it is important to understand and describe potential research variables that exists in inbred strains. To fully evaluate this disease in C57BL/6J mice, lesions were examined via computed tomography imaging, histologic examination, and mice underwent a variety of behavioral tests to determine the behavioral phenotypes of affected mice. Furthermore, with the inclusion of a morphine treatment group, we went on to determine if this disease is associated with pain in mice, or if their behavioral differences were purely due to their conformational change.

## Materials and methods

### Mice

Male C57BL/6J mice (JR 000664), aged 45–55 weeks, were bred and housed at The Jackson Laboratory in accordance with AAALAC international accreditation (accreditation number 00096) and the *Guide for the Care and Use of Laboratory Animals* standards (OLAW assurance D16-00170). All experimental procedures were approved by the Institutional Care and Use Committee (protocol number 99101) at The Jackson Laboratory. Mice were group housed (no more than 2 mice per cage) in micro isolation cages on individually ventilated racks (THO-REN Caging Systems, INC), with aspen shavings, with cage sanitation occurring on an every-2-week basis. Reverse osmosis, acidified water and a standard autoclaved rodent diet (Purina Lab Diet 5K52) were provided *ad libitum*. Routine health monitoring confirmed mice were free of viral antibodies, parasites, and opportunistic organisms [S1 Table]. Environmental conditions were maintained at 68 to 72°F (20 to 22°C) and 30%-70% humidity with a 12:12-h light: dark cycle (6:00am lights on: 6:00pm lights off). All experimental animals were humanely euthanized with carbon dioxide gas in accordance to the guidelines of the American Veterinary Medical Association.

### Experimental design

The colony of mice was observed closely in order to detect affected animals as soon as possible and affected and control mice were selected within a short window of time. A pilot study of 31 male mice, aged 45–55 weeks, were clinically evaluated to have a unilateral tarsal injury, evidenced by a rounded and thickened tarsal joint, abducted joint angle, and reduced range of

motion. A total of 70 age matched, unaffected male mice were identified for use as controls. Seven cohorts of affected mice with age matched non-affected mice underwent a battery of behavioral tests.

Frailty composite testing, voluntary wheel running assay, von Frey test, and the dynamic weight bearing test were conducted in that order, with at least 1-day rest in between tests. After behavioral testing a subset of mice were imaged via computed tomography and histopathology of affected and unaffected joints was evaluated.

After the pilot study, another group of male mice (n = 75), aged 45–55 weeks, that were clinically evaluated to have a unilateral tarsal injury and age matched, unaffected male control mice (n = 41) were selected for a follow up pain study. A battery of behavioral tests was performed without morphine: frailty composite testing, von Frey test, and dynamic weight bearing test, with at least 1-day rest in between. After a 7-day rest period between behavioral tests the groups were tested again with a 10 mg/kg dose of morphine sulfate in saline (Sigma, Lot# SLCG0429, Morphine sulfate pentahydrate, Cat. No. M8777-250mg, final concentration 1mg/mL) given via intraperitoneal injection approximately 1h before testing.

**Imaging.** A total of 10 affected and 13 unaffected age matched males were anesthetized with isoflurane and imaged on the Quantum GX Micro CT (PerkinElmer Health Sciences, Waltham, Massachusetts). All subjects had a single 72mm Field of View (FOV) whole-body scan at 288μm voxel resolution and a single 36mm FOV regional scan at 72μm voxel resolution of the hind limbs. Post scan, subjects were recovered in a clean 37˚C heated cage before returning to their home cages.

**Histopathology.** A total of 10 affected males were evaluated histologically in the affected tarsus and other joint surfaces. The tissue regions of interest were dissected and fixed in 10% neutral buffered formalin (Thermo Fischer Scientific, Fair Lawn, New Jersey) for 48 hours, and then decalcified in Immunocal (StatLab, Columbia, Maryland) for 96 hours. Tissues were then paraffin embedded, sectioned into 5 μm thick sections, and stained with hematoxylin and eosin. Sagittal sections of the tarsus and metatarsophalangeal joints were obtained. Multiple step sections were taken through the joints, ranging from 100 to 300μm between sections. Representative images were acquired using an Olympus BX41 microscope with an Olympus DP72 camera and cellSens Standard 1.5 imaging software. Histopathology evaluations were performed by a board-certified veterinary anatomic pathologist.

**Behavioral testing.** Before each behavioral procedure, mice were brought into the procedure room, body weights were recorded, and mice were left undisturbed to acclimate to the room for a minimum of 60 minutes. All tests were performed by trained and inter-validated technicians, as blind as possible using BlindID, and in a constant environment during the entire project [room temperature between 68 to 72˚F (20 to 22˚C)].

**Frailty composite test.** Mice were individually evaluated for the presence, absence, and severity of 27 categorical variables of wellbeing (scored 0, 0.5, or 1), as well as, body weight and body temperature (Braintree Scientific #TH5 Thermalert) [S2 Table]. Characteristics observed included physical, physiological and innate reflex conditions [4]. For frailty, the experimenters consisted of 4 different females, these were not repeated observations, and so each mouse was only observed by 1 female.

**Home cage wheel running.** Mice were individually placed in a clean cage equipped with a wireless running wheel (Med Associates, cat. no. ENV-047, components include DIG-807 USB interface hub, SOF-860 Wheel Manager Software, SOF-861 Wheel Analysis Software, and ENV-044-01 Low profile wheel) and the activity of the animals was recorded during a 2 day and 3-night session. Food and water were given *ad libitum* and daily checks performed. A wireless transponder, synchronized with a computer, recorded the number and time of

running wheel revolutions. Data was evaluated for onset of activity, distance traveled (rpm) and time spent running (min). For wheel running the experimenters were 2 different females.

**Von frey test.** Mice were acclimated for an additional 60-minute period on top of a wire mesh grid. One of a standard set of calibrated microfilaments (0.2 to 2g) was applied to the plantar surface of the hind paw until it bowed and was maintained in place for 3 seconds or until the paw was withdrawn. A positive response was noted as a withdrawal from the microfilament and response to the next lower microfilament were tested, consistent with the "up-down" method, where if no response was observed, then the next highest microfilament in the series is applied. On the highest filament the mouse is offered two opportunities to respond due to the up-down method, in which, once a mouse responds on the highest filament, they are presented with that filament again and if still a response is present the next lower filament is presented until the mouse does not respond to two filaments in a row [5]. For this test the 0.4g filament was the first filament applied in all mice. Both paws were tested subsequently and paw withdrawal threshold (g) is defined as the minimum force applied resulting in withdrawal of the paw. For von Frey, the experimenters were 3 different females.

**Dynamic weight bearing.** The dynamic weight bearing test (DWB) (BioSeb Dynamic Weight Bearing Instrument) was used to evaluate weight bearing and weight distribution per limb during unencumbered movement. It is an important component of motor abnormality assessment as it provides an underlying understanding of alterations in gait. Mice were weighed and individually placed in a chamber where they could move freely for a 5-minute period. The floor of the apparatus contains 1,936 pressure transducers, video from a high resolution camera mounted above the enclosure is used to corroborate data from the pressure transducers, and the raw data for each paw is synchronized with the images from the video to average the values of the paw weight and surface areas at a sampling rate of 10Hz [6]. For this test our data is based off an n = 64 for mice with tarsal lesions and n = 35 for control mice. This is in part due to not all mice providing the 10 seconds of no movement necessary for the equipment to obtain weight bearing movement. The entirety of the test lasts 300 seconds, in which time the machine needs 10 consecutive seconds of no movement to obtain data. Other part was due to scheduling conflicts and not being able to fit all mice into the testing schedule, resulting in some mice missing their before or after timepoints. In all we had 11 affected mice and 6 controls that we were not able to gather data from our original cohort of mice selected. Dynamic weight bearing experimenters contained 1 female and 1 male observer.

## Statistical analysis

Mixed model ANOVA analyses were conducted for the behavioral assays (SPSS software IBM v26) to account for possible cohort and age effects in the experiment. The bodyweight analysis was conducted only on the initial time point (before morphine). For other assays, group (Control vs Affected) was tested as a fixed factor, time-point (Before Morphine or After Morphine) as a repeated fixed factor, with cohort set as a random factor and age as a covariate. There were no significant effects of cohort. The Von Frey data also included the additional comparison of the two hind paws. The running wheel pilot study assay did not include the before/after morphine factor but had the additional fixed factor of light, the interaction of group with light and a repeated measure of the 30-minute binned measures embedded in the lighting condition.

## Results

### Physical examination

Grossly all affected mice presented with a tarsal joint abnormality in one hind leg, that generated an acute angle to the joint (compared to the normal near 90˚ angle)(red arrow), which

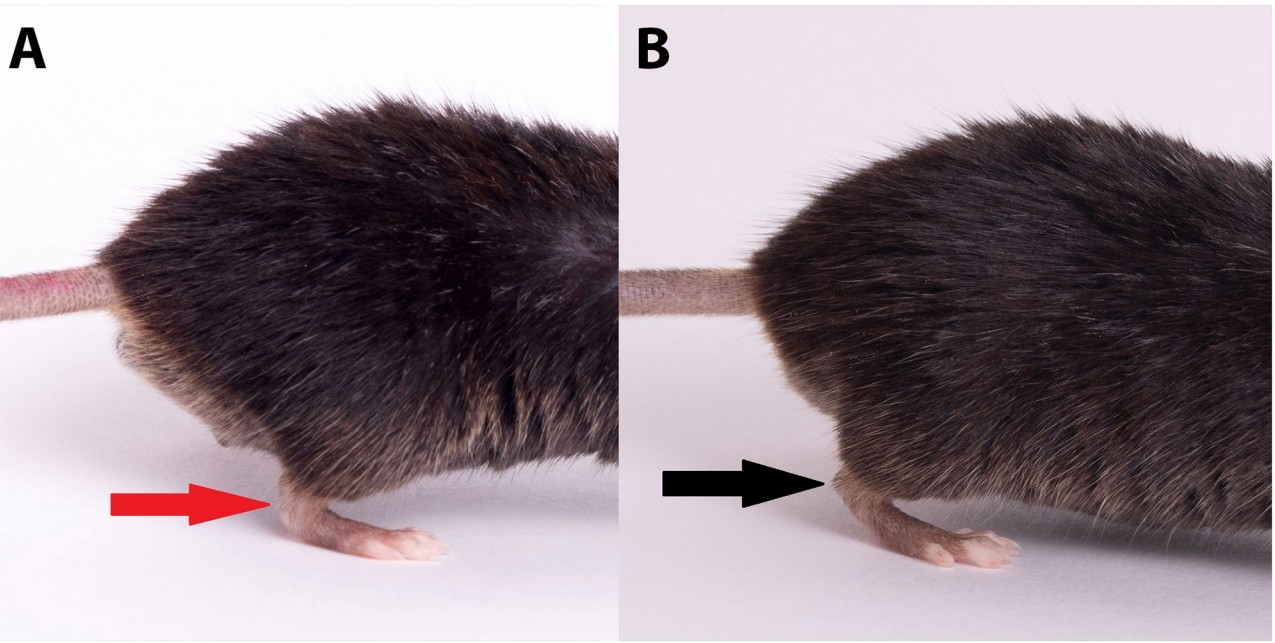

**Fig 1.** Lateral view of a male mouse with the right hind limb tarsal joint affected (A). Showing the acute angle of the joint, with a curved appearance (red arrow). Mice with this joint abnormality have an inability to fully extend the joint which results in them walking more on the metatarsals of the foot compared to a normal mouse (B) pictured on the right. Note the normal right hind limb joint, with normal positioning of the tarsal joint (black arrow) and weight bearing on foot pad.

was thick and stiff, inhibiting the ability to perform full extension of this joint (Fig 1A). This condition occurred in only 1% of the total population. The unaffected mouse is imaged (Fig 1B) and the normal near 90 angle to the joint is highlighted by the black arrow. Note how the affected mouse (Fig 1A) has more of the metatarsals planted on the ground compared to an unaffected mouse (Fig 1B) that has more of the phalanges of the feet touching the ground with the metatarsals lifted off the ground.

## Imaging

MicroCT in all affected mice revealed an acute angle to the tarsal joint, with severe smoothly marginated new bone at entheses and bridging the joint margins (Fig 2B, 2D and 2E), compared to unaffected mice (Fig 2A and 2C). The joint spaces were narrowed and ill-defined, and the margins of the tarsal bones were indistinct from each other (Fig 2E). The regional soft tissues surrounding the tarsus were circumferentially thickened (Fig 2D solid arrows). Imaging findings were consistent with dislocation of the calcaneus and tarsal ankylosis. Thin, linear mineralization was within the calcaneal tendon of both unaffected and affected mice (Fig 2A–2D).

## Histopathology

Histopathological examination identified luxation and proximocaudal displacement of the calcaneus, such that the long axis of the calcaneus was nearly parallel to the long axis of the tibia, in all grossly abnormal joints (Fig 3B), when compared to normal orientation in unaffected joints (Fig 3A). There was often partial to complete osteonecrosis of the talus (Fig 3C), which also had histological evidence of subluxation (Fig 3C) in some of the mice. Bone remodeling of

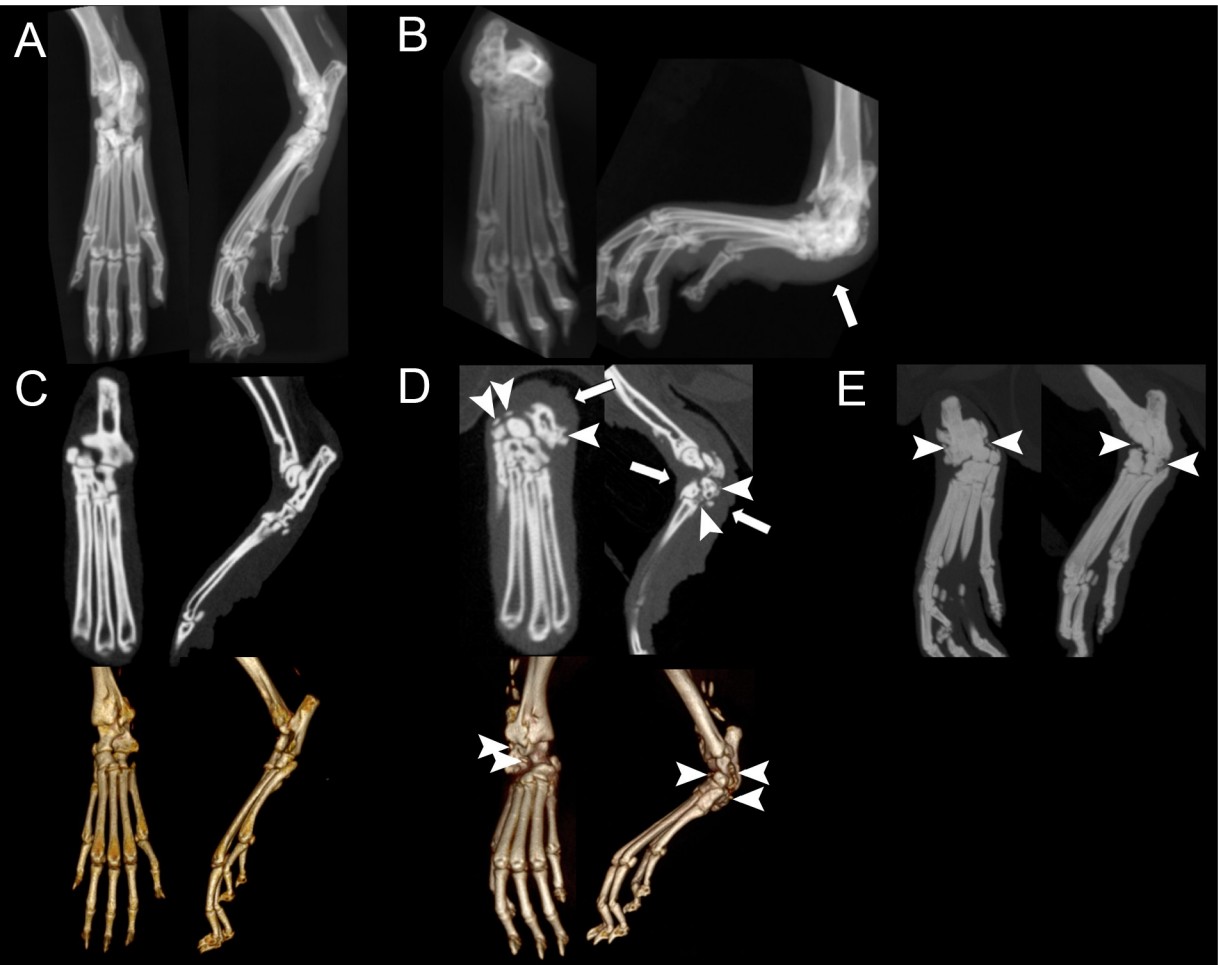

**Fig 2.** Dorsoplantar (left) and lateral (right) radiographs of an unaffected tarsal joint (A) and an abnormal, affected tarsal joint (B). Coronal (left) and sagittal (right) microCT manual planar reformatted images (top) and 3D reconstructed images (bottom) of an unaffected tarsal joint (C) and an abnormal, affected tarsal joint (D). Coronal (left) and sagittal (right) maximum intensity projection images of an abnormal, affected tarsal joint (E). Note the acute angle to the affected tarsal joint. There is a large volume of smoothly marginated mineralization along the periarticular margins of the tarsus including entheses of the distal tibia, tarsal joints, and proximal metatarsus, bridging the joint spaces and tarsal bones (dashed arrows). The soft tissues centered over the affected tarsus were circumferentially thickened, worse plantar (white arrows).

the tarsal bones was marked, with periosteal proliferation, large enthesophytes (Fig 3B, 3E and 3G open arrows), osteophytes, cartilaginous to boney metaplasia of the soft tissues (Fig 3E and 3G open arrows), and partial ankylosis. Most affected joints had moderate synovial hyperplasia and mild inflammation and fibrovascular proliferation (Fig 3E and 3G). The articular cartilage between tarsal bones was often smooth or had only mild fibrillation (Fig 3B and 3C evidenced with closed arrows). The overlying skin was usually normal, with no inflammation or ulceration noted (Fig 3B and 3C).

## Frailty assessment

The frailty score was calculated as a sum of all sub-scores attributed in several observational, physical, and physiological features. Data are means ± standard deviation. Mice affected by unilateral calcaneal displacement had a significantly lower body weight (mean = 36.03g ± 0.425) compared to controls (38.41g ± 0.575) (p<0.001 age significant

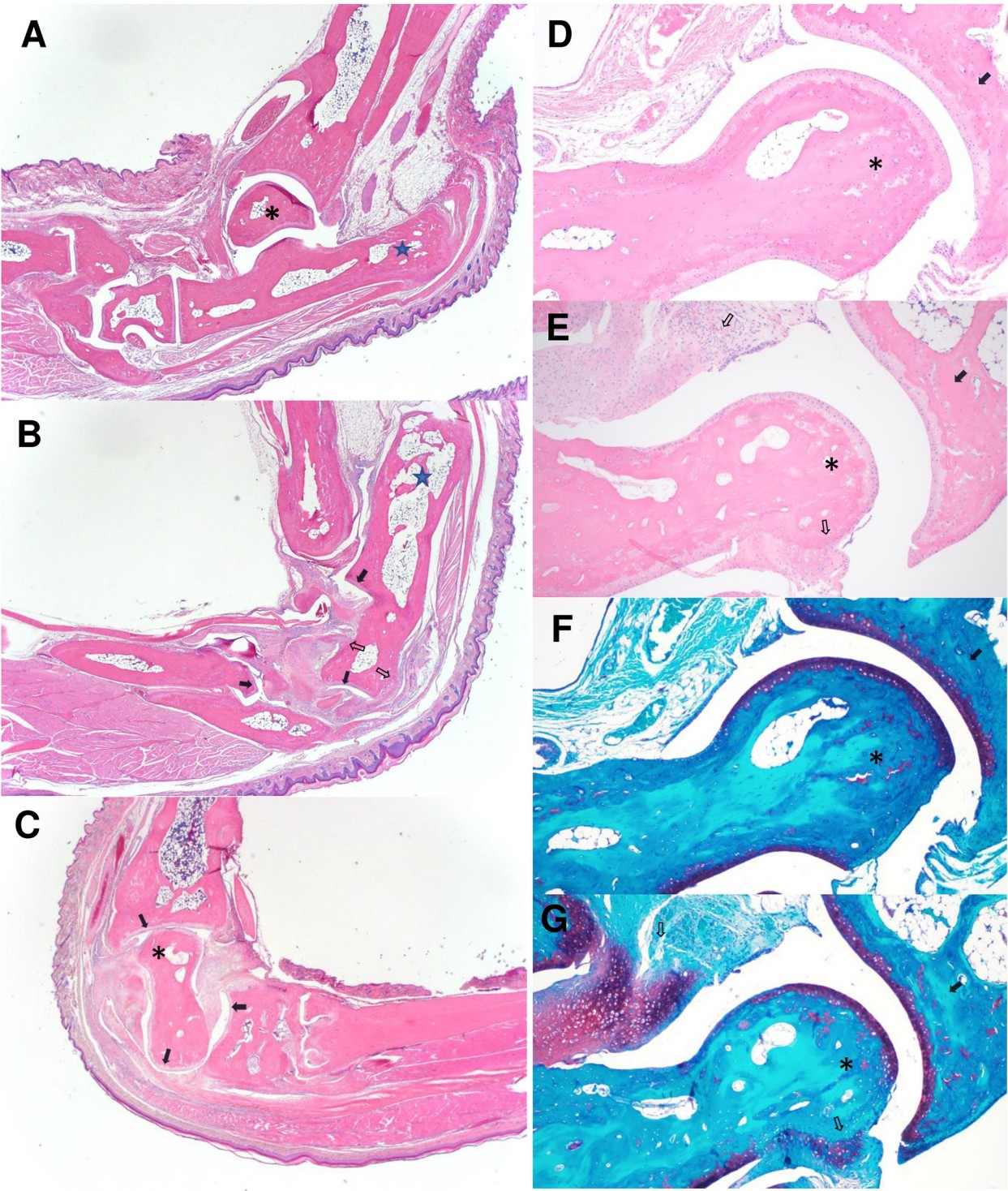

**Fig 3.** Lateral view of the tibiotarsal joint of an unaffected paw (A) and affected paws showing displacement of the calcaneus (B) and the talus (C) bones. The calcaneus (star) is displaced proximally and caudally in the affected paw with the long axis nearly parallel to the tibia. The talus (*) of the affected paw (C) is necrotic and the talotarsal joint is subluxated with the talar body rotated cranially and the talar head rotated caudally. Affected joints have extensive fibrocartilaginous proliferation of the joint capsule and ligaments (open arrows) resulting in partial ankylosis. Note that articular cartilage is often essentially normal (solid arrows). Higher magnification views of the tibiotalal joint of unaffected (D, F) and affected (E, G) paws using H&E (D, E) and Safranin O (F, G) stains.

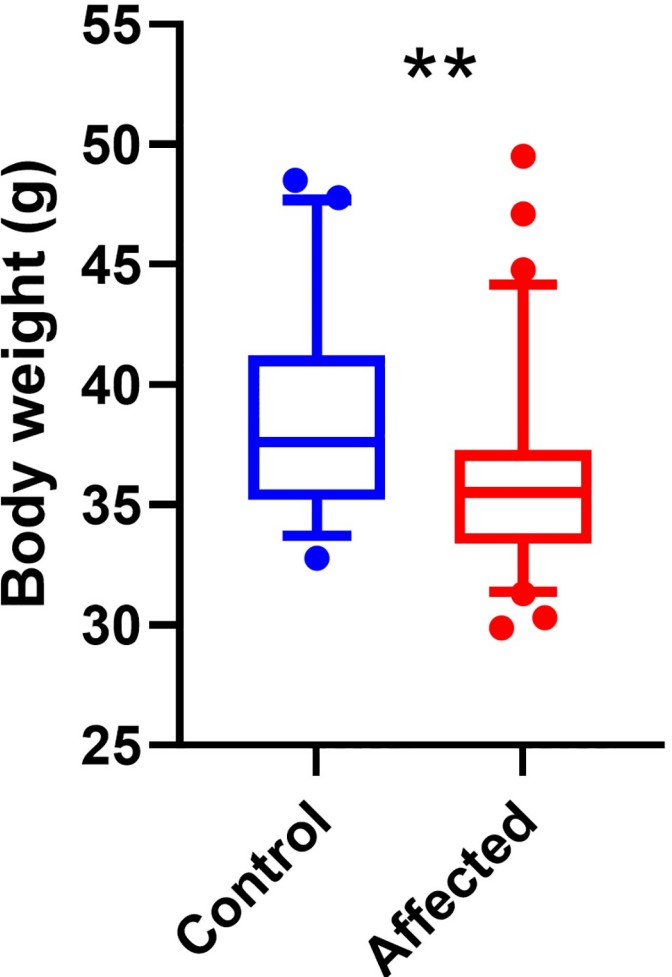

**Fig 4. Bodyweights for the control group (n = 41) and affected group (n = 75), measured at the before-morphine time-point, are displayed as boxplots, showing mean, quartiles with 5–95 percentile error bars.** (** p<0.01).

covariate p<0.01; $F_{1,113}$ = 11.1.8, p<0.01) (tested before morphine) (Fig 4). Affected mice displayed a significant increase in frailty score (p<0.05) compared to control mice both before (Affected mean = 5.4 ± 0.2; Control 4.9 ± 0.3; $F_{1,113}$ = 6.7, p<0.05) and after administration of morphine (Affected 5.3 ± 0.2, Control 4.6 ± 0.3; $F_{1,113}$ = 19.5, p<0.001) (Fig 5). The results of the pilot data were similar, as affected mice had lower body weights (Affected 36.2 ± 3.7; Control 39.3 ± 4.8; $F_{1,97}$ = 17.1, p<0.001) and had higher frailty scores (Affected 4.5 ± 0.8; Control 4.1 ± 1.1; $F_{1,96}$ = 3.9, p = 0.05) compared to control mice.

## Home cage wheel running

Home cage wheel running was only performed during the pilot study. Running wheel data was averaged in 30-minute bins and pooled across days in each light condition representing a 24-hour cycle. The total duration of the running wheel experiment was a 2 day and 3 night session. The mean distance travelled and mean time on wheel (in 30 minutes) were recorded for 12 hours of dark and 12 hours of light as 24 bins in each light condition. The mean overall distance travelled by the affected mice was greater than the control mice ($F_{1,11014}$ = 6.9, p<0.015).

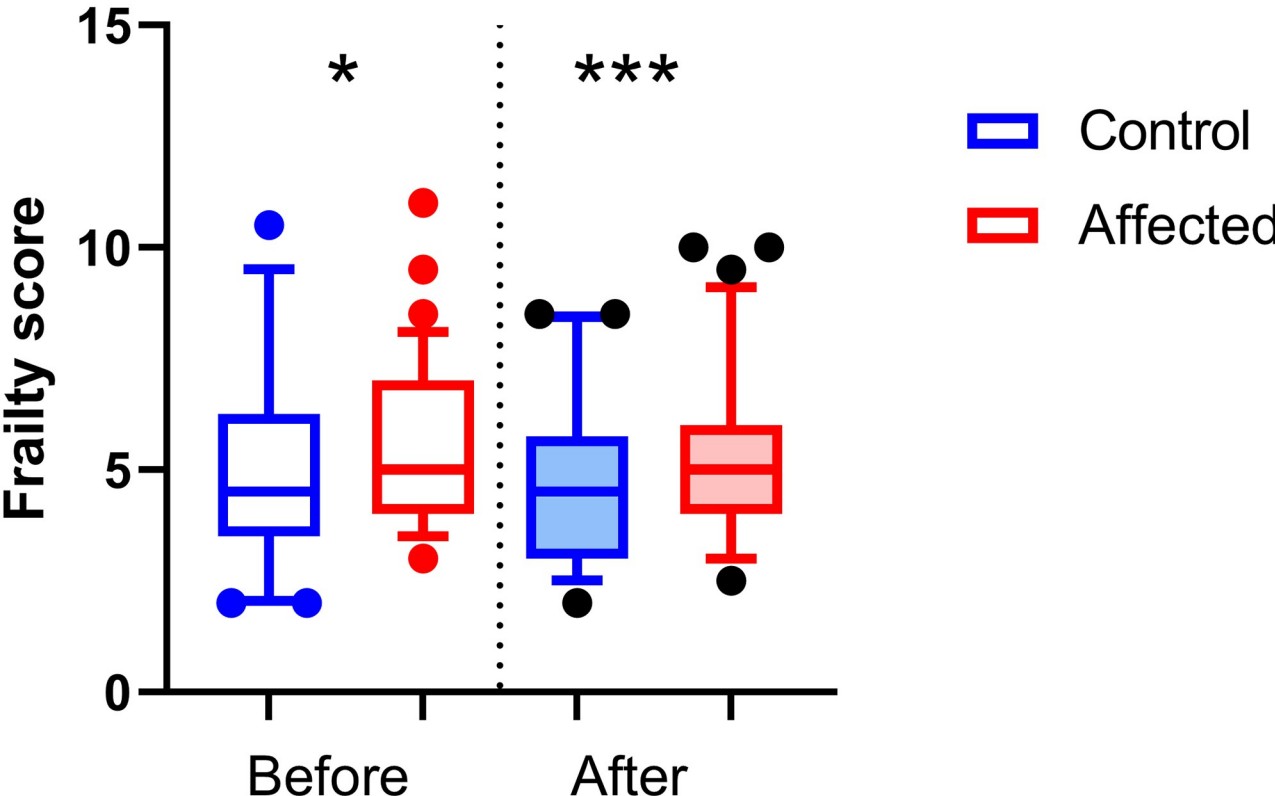

**Fig 5. The frailty scores for the control group (n = 41) and affected group (n = 75) displayed as boxplots, showing mean, quartiles with 5–95 percentile error bars before (\*p<0.05) and after (\*\*\*p<0.001) morphine.** Data are means ± SD.

More specifically, the interaction of factors revealed that the distance travelled in the dark was significantly greater for the affected group compared to control mice ($F_{1,11014}$ = 8.7, p<0.01) with no significant differences found in the light. Fig 6A shows that affected mice travelled further in the dark period except for the few bins between 4:30 and 6:00am. In contrast, mean time on the wheel during the dark was significantly lower for affected animals, ($F_{1,11008}$ = 7.6, p<0.01) than controls with no differences seen in the light. Fig 6B shows that the difference between the groups was more pronounced in the hours before and after light transition.

### Von frey

To test for hind foot withdrawal threshold, affected mice had both affected and nonaffected feet tested, and control mice had both right and left hind limbs tested. Data was log transformed prior to analysis. Factors of group (Affected vs Control), time (Before and After morphine) and foot (affected vs non affected for Affected and left vs right for Control) were tested. It was predicted that affected would show differences in sensitivity between the feet and the control group would not. This was shown as a significant interaction between group and foot ($F_{2,468}$ = 9.99, p<0.001). Affected mice showed lowest response threshold on their affected hind limb, with their unaffected limb having a similar threshold to control mice (Fig 7A). The groups also responded differently to morphine ($F_{2,468}$ = 6.5, p<0.01). After morphine, the affected mice response threshold increased slightly, but still had a lower response threshold than their unaffected limb or compared to the control mice (Fig 7B).

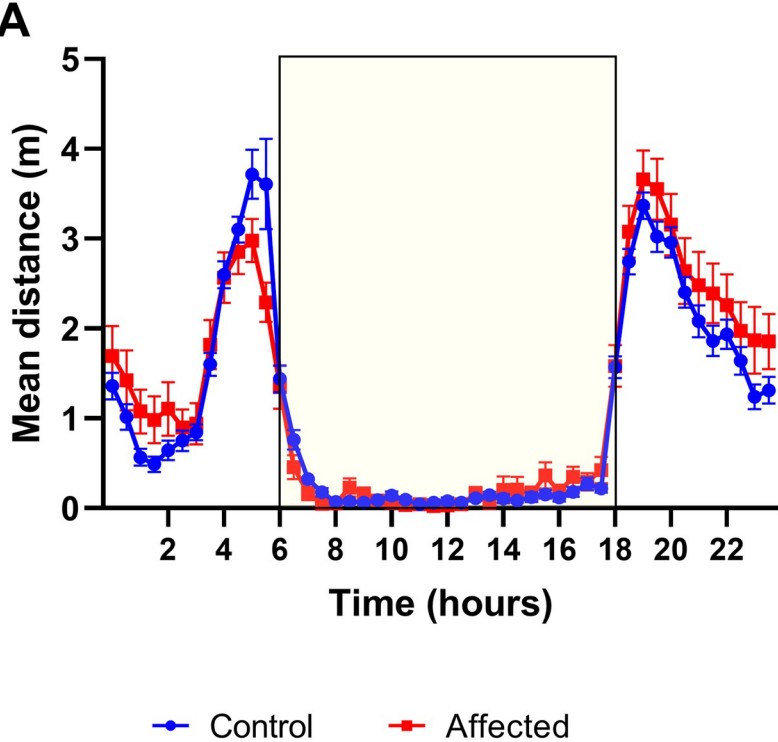

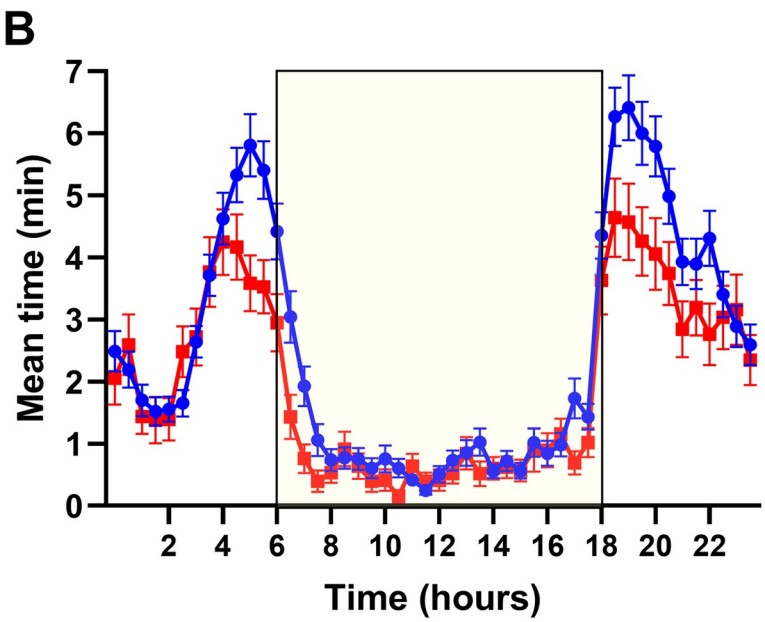

**Fig 6.** A) The mean distance travelled (+/- SEM) for each 30-minute bin is displayed in a 24-hour cycle for the control (n = 70) and affected (n = 31) mice. The yellow box indicates lights on. B) The mean time on the wheel in minutes (+/- SEM) for each 30-minute bin is displayed in a 24-hour cycle (yellow box indicates lights on).

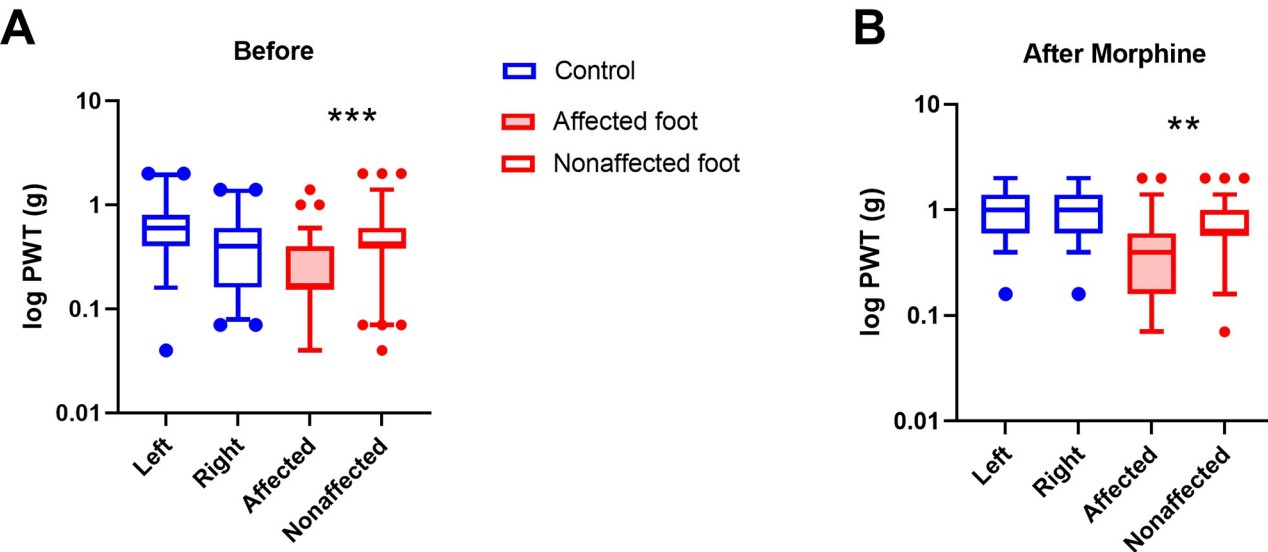

**Fig 7.** A) The log transformed force of both hind limbs before morphine administration for the control group (n = 41), and both affected and unaffected hind limbs of the affected group (n = 75) are displayed as boxplots, showing mean, quartiles with 5–95 percentile error bars (***p<0.001). B) The threshold force of both hind limbs after morphine administration for the control group (n = 41), and both affected and unaffected hind limbs of the affected group (n = 75) (**p<0.01).

### Dynamic weight bearing

To test for differences in weight bearing, the stance of mice was examined by comparing the load placed on forelimbs and hind limbs. Mean surface area and mean weight load were determined for fore and hind limbs and normalized to body weight. The analysis compared the factors of stance (Fore vs Hind limbs), group (Affected vs Control), and time-point (Before vs after morphine). It was predicted that affected mice would place relatively less load on their hind limbs compared to control mice and this would be shown as an interaction between the factors of stance and group.

The surface area interactions were significant between stance and group ($F_{1,96}$ = 6.3, p<0.05) and between time, stance, and group ($F_{1,96}$ = 5.0, p<0.05). The hind limb surface area before morphine for the affected group was lower (mean = 0.57 ± 0.12 mm$^2$) compared to control mice (0.66 ± 0.11; Univariate ANOVA post hoc comparison $F_{1,96}$ = 13.4, p<0.001), while the forelimb surface area was similar to control mice (Affected 0.33 ± 0.11; Control 0.32 ± 0.11; Fig 8A). However, after morphine, there was no significant difference between the affected (0.53 ± 0.11) and control groups hind limb surface area (0.52 ± 0.09; Fig 8B).

The normalized weight load interactions were significant between stance and group ($F_{1,96}$ = 4.1, p<0.05) and between stance and time ($F_{1,96}$ = 12.0, p<0.001). Before morphine, the proportion of the weight load placed on the hind limbs of the affected group (mean = 0.66 ± 0.12), was lower compared to control mice (0.69 ± 0.10; Univariate ANOVA post hoc comparison $F_{1,96}$ = 4.0, p<0.05; Fig 9A) After morphine, the affected group hind limb weight load proportion increases (0.69 ± 0.12) and is barely lower than the control group (0.71 ± 0.12; Univariate ANOVA post hoc comparison $F_{1,96}$ = 3.9, p = 0.05; Fig 9B).

### Discussion

This study sought to phenotype mice with unilateral tarsal abnormalities to better understand the potential confound this condition introduces for research subjects and determine if this

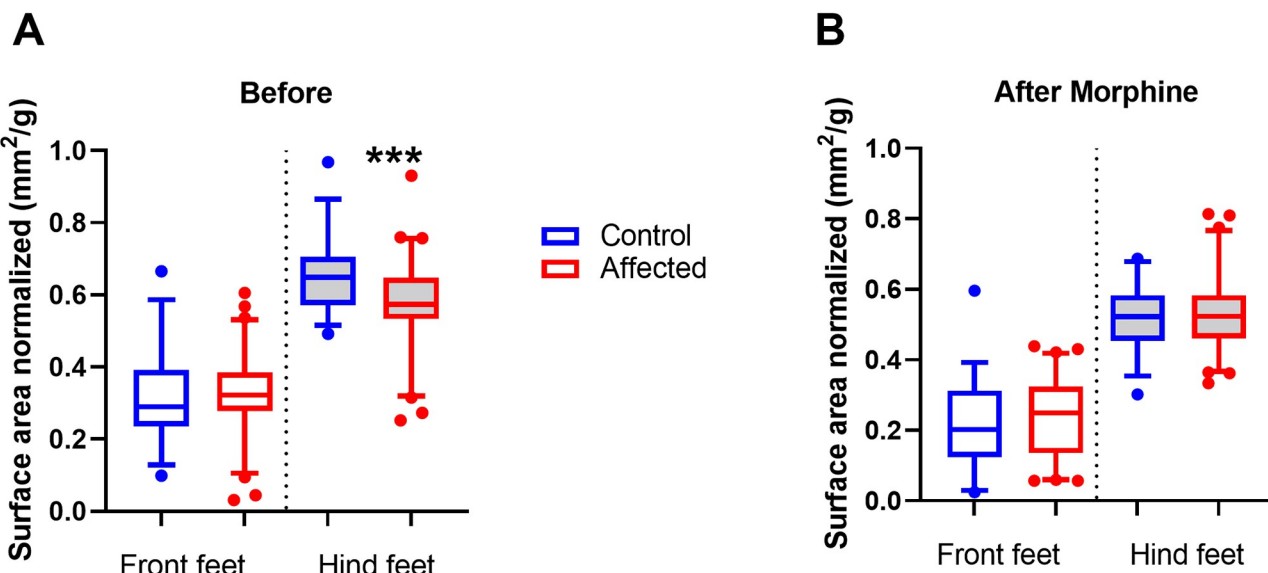

**Fig 8.** A) Before morphine surface area fore and hind limb scores for control group (n = 35) and affected group (n = 64) displayed as boxplots, showing mean, quartiles and 5–95 percentile error bars (***p<0.001) 8B). After morphine surface area fore and hind limb scores for control group (n = 35) and affected group (n = 64).

condition results in nociception. Overall, we only observed this condition in older male mice of the C57BL/6J lineage, around the age of 20 weeks or older, at a frequency of 1% of the population. Moreover, this study demonstrates that mice with unilateral tarsal abnormalities have higher frailty scores, weigh less, bear more weight on their unaffected limbs, have higher touch

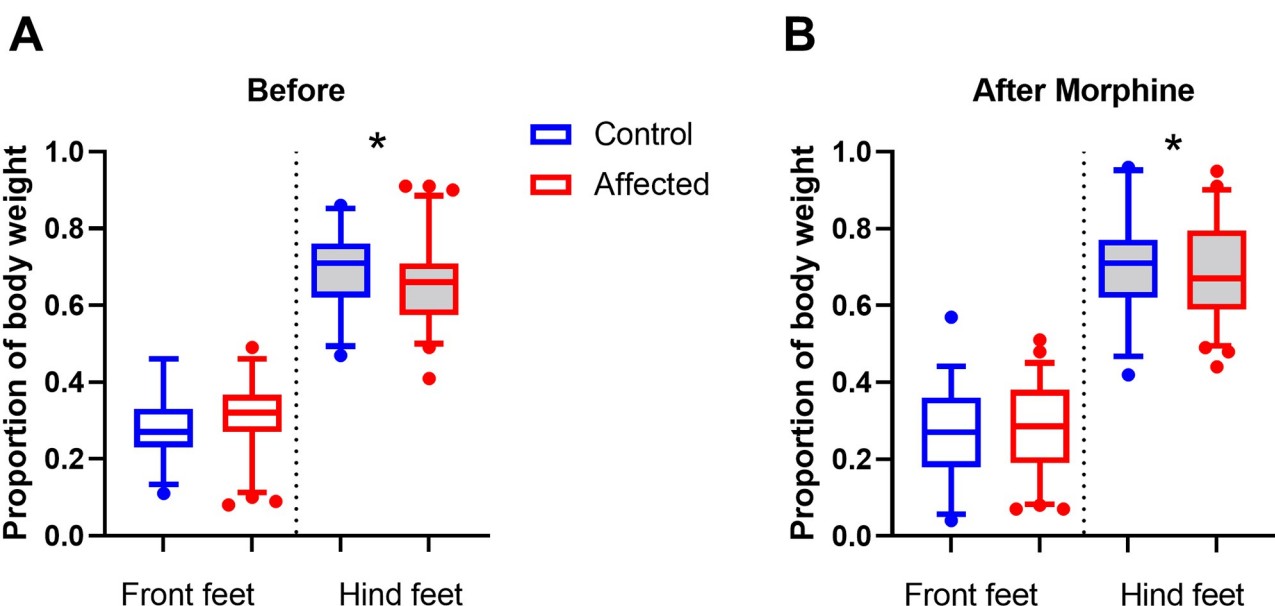

**Fig 9.** A) Before morphine normalized weight load score for both fore and hind limbs for control group (n = 35) and affected group (n = 64) displayed as boxplots, showing mean, quartiles and 5–95 percentile error bars (p<0.05) B). After morphine normalized weight load score for both fore and hind limbs for control group (n = 35) and affected group (n = 64) (*p = 0.05).

sensitivity on their affected limb, and run farther in a shorter amount of time compared to unaffected mice. The morphine data suggests that affected mice do experience nociception with this condition, as after morphine treatment, mice were able to bear more weight on their affected limb and their touch sensitivity decreased with the von Frey filaments.

During the behavioral tests with morphine administration our affected mice still did not respond equivalently to our control groups and some avenues of thought include that either their bone structure confirmation difference does affect their behavior, the morphine dose was not at an effective dose for these mice to completely ameliorate nociception, and/or the time window in which we performed the behavioral tests was not within the optimum timeframe for maximum effectiveness. We did utilize a published dose of morphine and stayed within the published effective timeframe, however, morphine doses in mice were published in the ICR mouse strain and strain differences are noted with other drug administrations in mice [7]. Their bone conformation changes likely play a role in some results, such as dynamic weight bearing. Since we measured surface area, we know that the bone conformation causes them to have less surface area on their affected limb. This is why we wanted to evaluate the difference between fore and hind limbs during evaluation so that we could average out their affected and non-affected limbs in the results, compared to most of our data where we utilized their unaffected hind limb as an internal control. Memory may have also played a role in some of the behavioral responses. For instance, with von Frey, while the affected mice showed a decrease in touch sensitivity post morphine administration, they still did not respond equivalently to control mice, which may be due to remembering the prior event and the mechanical nociception that it caused, resulting in an increased touch sensitivity to control mice with morphine. Overall, there were improvements seen in the behavioral responses in the tests we performed, which is a strong indicator that nociception is a sensation for the mice with tarsal abnormalities.

All of the affected mice were at a chronic stage of pathology, so it is not clear how the tarsal pathology started. However, since there was no evidence of pathology in the unaffected joints, the dislocation of the calcaneus may be the initial event, and the resulting joint instability leads to connective tissue and cartilage proliferation, inevitably ending with ankyloses of the joint. The pathologic changes, along with higher frailty scores, lower body weight, lower mean run time during the dark phase, lower response threshold to von Frey filaments, and less weight placed on affected limbs in dynamic weight bearing indicate that mice are unable to respond to various tests in a similar manner as unaffected controls. With the results from the morphine data, in which affected mice showed improvement in their von Frey scores and more evenly distributed weight on dynamic weight bearing is suggestive of this condition having nociceptive properties. With C57BL/6J mice being a common background strain to utilize in genetic manipulations, it is important to identify and characterize any background pathology. This disease may be an unwanted confounding research variable depending on research goals, and therefore, the condition may need to be identified and removed from the research cohorts.

While the exact cause of the tarsal abnormality remains unknown, there are some similarities with other joint diseases noted predominantly in males. Male mice have a higher incidence of osteoarthritis (OA), particularly in the STR/ort strain, and this strain has been also observed to have calcaneal dislocation [3, 8, 9]. Males are also exclusively affected by Ankylosing Enthesopathy (ANKENT), a spontaneous joint disease that begins with development of inflammatory cells at enthesis sites, followed by proliferation of cartilage and connective tissue, and results in or causes ankyloses of the tarsal joint [2, 10, 11]. Similar to the C57BL/6J mice affected in this study, ANKENT occurs in group housed male mice on a C57BL background (most notably the B10.BR strain), and does not develop in singly housed males, breeder males, or females of a similar age range [12, 13]. While OA and ANKENT are two different

musculoskeletal disease processes, they are both sexually dimorphic in which similar strains and ages of female mice do not develop these disease, however, it has been noted in both diseases that there is no linkage sex hormones [12]. Group housed males are known to have higher bone mineral density and content, which may be due to their incidence of fighting being 1.3 times higher than group housed females [14, 15]. The increased fighting is likely putting enough strain on the hindlimbs to prompt an osteogenic response [14]. Notably, our microCT scans showed that even in nonaffected aged males, the calcaneal tendons contained small fragments of bone, which may be a result of chronic rearing injuries during fighting events or a result of normal aging degenerative changes. This change was noted as an apparent normal degenerative aging process in both sexes of STR/ort and CBA/ort strains, for mice over the age of 7 months [3].

While the exact cause for this tarsal abnormality remains unknown, it is likely to have a multifactorial cause, similar to ANKENT, in which a combination of genetics, housing condition, and microbial flora all play a role in the disease onset and progression [10–13, 16]. An advanced understanding of how tarsal abnormalities affect various behavioral/phenotypic tests will help other institutions or research groups identify this condition and understand that this disease may result in research variables. While in our facility the rate of occurrence in our aged C57BL/6J colony was around 1%, other institutions have reported as high as 12% occurrence dependent on strain and other genetic manipulations, suggesting the important role of the environmental parameters in the development of the disease [1]. Further research to fully understand how this disease develops, and how to prevent it, will help preserve research subjects in line with the 3 R's of Russell and Birch [17].

## Supporting information

**S1 Table.**
(PDF)

**S2 Table.**
(DOCX)

## Acknowledgments

The author would like to thank The Jackson Laboratory Center for Biometric Analysis, Scientific Services, and Comparative Medicine and Quality staff for all of their contributions to this body of work. A special thank you to Jacqui White for all of her financial and review support.

## Author Contributions

**Conceptualization:** Brenda L. Kick, Rosalinda Doty, Bonnie Lyons, Zoë Bichler.

**Data curation:** Laura Anderson, Rosalinda Doty, Christine Wooley, Meaghan Dyer, Torrian Green, Veronica Knickerbocker, Zoe Brown, Samantha Loeber, Zoë Bichler.

**Formal analysis:** Rosalinda Doty, Samantha Loeber, Janine Wotton, Zoë Bichler.

**Funding acquisition:** Linda Waterman.

**Investigation:** Brenda L. Kick, Zoë Bichler.

**Methodology:** Brenda L. Kick, Bonnie Lyons.

**Project administration:** Brenda L. Kick.

**Resources:** Laura Anderson, Janine Wotton, Bonnie Lyons, Linda Waterman.

**Software:** Samantha Loeber, Janine Wotton.

**Supervision:** Brenda L. Kick, Bonnie Lyons, Linda Waterman.

**Validation:** Janine Wotton.

**Writing – original draft:** Brenda L. Kick, Janine Wotton.

**Writing – review & editing:** Laura Anderson, Rosalinda Doty, Samantha Loeber, Bonnie Lyons, Zoë Bichler.

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
