## [Decision Letter · Decision Letter 0]

19 Dec 2022

PONE-D-22-26026Evaluation of Tarsal Injuries in C57BL/6J Male MicePLOS ONE

Dear Dr. Kick,

Thank you for submitting your manuscript to PLOS ONE. After careful consideration, we feel that it has merit but does not fully meet PLOS ONE’s publication criteria as it currently stands. Therefore, we invite you to submit a revised version of the manuscript that addresses the points raised during the review process.

Specifically, both reviewers raised significant concerns that need your attention. Please have all to comments addressed. 

We look forward to receiving your revised manuscript.

Kind regards,

Jianhong Zhou

Staff Editor

PLOS ONE

Journal Requirements:

Reviewers' comments:

Reviewer's Responses to Questions

**Comments to the Author**

1. Is the manuscript technically sound, and do the data support the conclusions?

Reviewer #1: Partly

Reviewer #2: Partly

2. Has the statistical analysis been performed appropriately and rigorously? 

Reviewer #1: No

Reviewer #2: Yes

3. Have the authors made all data underlying the findings in their manuscript fully available?

Reviewer #1: No

Reviewer #2: Yes

4. Is the manuscript presented in an intelligible fashion and written in standard English?

Reviewer #1: Yes

Reviewer #2: Yes

5. Review Comments to the Author

Reviewer #1: November 15, 2022

Plos One Manuscript Review

Title: Evaluation of Tarsal Injuries in C57BL/6J Male Mice (PONE-D-22-26026)

Authors: Brenda Kick et al.

Summary and Comments: In this manuscript, the authors expand on previous reports of unilateral tarsal abnormalities found in a small subset of C57BL/6N, STR/ort, and B10.BR mice. The authors report that 1% of C57BL/6J male mice similarly display tarsal abnormalities by microCT and histological analyses. Furthermore, the authors go on to show that affected mice are responsive to morphine with regards to motor performance, suggesting that they experience pain as a result of tarsal abnormalities. Interestingly, these features were not observed in females or individually housed male mice indicating that they may be associated with unique behavior in male mice.

Overall, this study provides additional information on a subpopulation of mouse strains with spontaneous unilateral tarsal abnormalities, particularly in older C57BL/6J male mice which may impact data interpretation when using this mouse strain. One of the main concerns is that the manuscript appears to be incompletely written in its current form as the figures are not fully described in the text throughout the results section. In addition, it is unclear whether the phenotypes described are representative as there is little quantification of the morphological or histopathological data. Lastly, the manuscript should clearly state in the abstract and discussion (not just the introduction) that the unilateral tarsal abnormalities found in C57BL/6J mice are found only in a small portion of animals. The reviewer recommends that these issues be addressed prior to consideration for publication.

Other Comments:

In both the abstract and results section, it should be clearly stated that the spontaneous tarsal aberrations that the authors are describing are observed only a small population of mice, similar to what is stated in Line 62-63 (“Abnormal tarsal joints were noted in older male mice of the C57BL/6J lineage, around the age of 20 weeks or older, at a frequency of 1% of the population”).

Line 56, Authors’ description of the tarsal abnormalities, “ranging from 1.7% up to 12%, and beginning from 4 weeks of age to 78 weeks” seems very broad and random. If more information is available, it would be helpful to the reader if the authors discussed whether the rate of tarsal abnormalities are associated with age, sex, etc.

Line 127-128, it is mentioned that histological sections for the lumbar spine, coxofemoral joint, and TMJ were generated but the data is nowhere to be found. Either include the data or remove the description in the Methods section.

Line 138, more details on the Frailty composite testing method is needed to help readers understand how the experiment was actually performed.

In Fig 1, Readers will find it helpful if the hindfoot joint abnormality is indicated with an arrow or outlined to show the extent of the curvature.

Line 199, Do the authors have data to demonstrate that “Mice with this joint abnormality have an inability to fully extend the joint…walking more on the metatarsals of the foot”?

In Fig 2, dotted arrows should be edited to enhance visibility.

Line 205-210, There is no mention of Fig 2A or 2C-E in the results section. Also applicable to other figures throughout the manuscript.

Line 227-230, histological description is difficult to confirm due to the low resolution of images presented. Changes in proliferation, osteophyte formation, synovial inflammation should be quantified or assessed by histological scoring accompanied by statistical analysis.

Line 234, asterisk is marking the talus in the actual figure. The star is marking calcaneus but is not described in the figure legend.

Line 239, higher resolution images, proper staining (e.g., Safranin-O, etc.) and quantification are needed to show the presence or absence of damage to the articular cartilage.

Line 363-366, there is no basis for the claim that calcaneal dislocation is the underlying cause of tarsal pathology.

Reviewer #2: This interesting study explores the occurrence of tarsal injuries in the C57BL/6J mouse strain (males only). While the initial characterization provides exciting data that corroborates some of the authors' claims, additional analyses are needed to reach the conclusions presented in the manuscript.

First, histological features highlighted in the text, including tarsal necrosis, synovial hyperplasia, periosteal proliferation, osteophytes, and fibrocartilagenous formations within ligaments, are poorly captured in the Figures in large part due to a lack of appropriate histological staining and high-magnification images. At a minimum, the authors should utilize stains that evaluate the integrity of the ECM, particularly in synovial compartments, ligaments and tendons of the foot (including Achilles tendon), bone, and cartilage. Stains such as safranin-O/fast green or toluidine blue for cartilage, picosirius red to evaluate collagen alignment by polarized light microscopy, and Masson's trichrome should be performed. Sections that include the Achilles tendon should also be shown in addition to the ones provided, and higher magnification images of the site of dislocation, especially the areas of aberrant ECM deposition seen adjacent to the calcaneal displacement, should be included. Low and high-res photos should be provided side-by-side for normal and affected mice.

The available microCT data provided is compelling but under-analyzed. In particular, could joint space narrowing be quantified and/or bone mineral density of affected bones in the foot? Also, how is the shape of the bones affected compared to mice without the deformity - could this be quantified in some way?

The use of a frailty score comprising 27 individual observational scores is uninformative. The authors should present these 27 scores individually in a Table to accompany the combined Frailty Score given in Figure 4, along with the appropriate statistical tests for each. In addition, how they were collected, including room conditions, the gender of the observer, whether the observer was blinded, and whether multiple observers were used, should be included in the Methods section. This type of information should be added for all behavioural assays used.

Overall, the behavioural data is interesting, and the pain contribution is compelling based on the analysis of Morphine administration. However, have the authors evaluated whether the innervation of the foot is altered? Or is inflammation elevated in affected animals?

6. PLOS authors have the option to publish the peer review history of their article (what does this mean?). If published, this will include your full peer review and any attached files.

Reviewer #1: No

Reviewer #2: No

---

## [Author Response · Author response to Decision Letter 0]

19 May 2023

Comments:

In both the abstract and results section, it should be clearly stated that the spontaneous tarsal aberrations that the authors are describing are observed only a small population of mice, similar to what is stated in Line 62-63 (“Abnormal tarsal joints were noted in older male mice of the C57BL/6J lineage, around the age of 20 weeks or older, at a frequency of 1% of the population”).

Please note updated lines 31-33, 209-210, and 363-364 to address this concern. 

Line 56, Authors’ description of the tarsal abnormalities, “ranging from 1.7% up to 12%, and beginning from 4 weeks of age to 78 weeks” seems very broad and random. If more information is available, it would be helpful to the reader if the authors discussed whether the rate of tarsal abnormalities are associated with age, sex, etc.

The exact reasoning for the broad and random differences that were noted within the International Mouse Phenotyping Consortium is unknown. It was hypothesized in the paper that potentially it may be due to animal housing and husbandry practice differences. I added further clarification in the text and potential other hypotheses to why the difference exists in lines 61-66. 

Line 127-128, it is mentioned that histological sections for the lumbar spine, coxofemoral joint, and TMJ were generated but the data is nowhere to be found. Either include the data or remove the description in the Methods section.

Description removed from Methods section 

Line 138, more details on the Frailty composite testing method is needed to help readers understand how the experiment was actually performed.

Please see additional Supplement 2 Table provided to the readers. 

In Fig 1, Readers will find it helpful if the hindfoot joint abnormality is indicated with an arrow or outlined to show the extent of the curvature.

Please note that there are now red and black arrows highlighting the hind foot joint abnormality. 

Line 199, Do the authors have data to demonstrate that “Mice with this joint abnormality have an inability to fully extend the joint…walking more on the metatarsals of the foot”?

This was a clinically observed phenomenon, no data exists. Figure 1 does show that the metatarsals are more planted on the ground compared to the normal mouse in the image. Please see further description in results lines 209-214. 

In Fig 2, dotted arrows should be edited to enhance visibility.

Line 205-210, There is no mention of Fig 2A or 2C-E in the results section. Also applicable to other figures throughout the manuscript.

Corrected in the respective results sections lines 224-231. 

Line 227-230, histological description is difficult to confirm due to the low resolution of images presented. Changes in proliferation, osteophyte formation, synovial inflammation should be quantified or assessed by histological scoring accompanied by statistical analysis.

See updated Figure 3. Note we did not pursue a histologic scoring mechanism as most of the grading systems out there are for the knee (one large joint) compared to the tarsus (multiple small joints). There is also not a good view of all of the small joints in the tarsus due to the function of the cut, making scoring difficult. 

Line 234, asterisk is marking the talus in the actual figure. The star is marking calcaneus but is not described in the figure legend.

Corrected in lines 260-261

Line 239, higher resolution images, proper staining (e.g., Safranin-O, etc.) and quantification are needed to show the presence or absence of damage to the articular cartilage.

See updated image 3. We added views with Safranin-O stain. We didn’t get to show images that we obtained with Trichrome stains due to folds being present in the cut. 

Line 363-366, there is no basis for the claim that calcaneal dislocation is the underlying cause of tarsal pathology.

See updated line 393.

Reviewer #2: This interesting study explores the occurrence of tarsal injuries in the C57BL/6J mouse strain (males only). While the initial characterization provides exciting data that corroborates some of the authors' claims, additional analyses are needed to reach the conclusions presented in the manuscript.

First, histological features highlighted in the text, including tarsal necrosis, synovial hyperplasia, periosteal proliferation, osteophytes, and fibrocartilagenous formations within ligaments, are poorly captured in the Figures in large part due to a lack of appropriate histological staining and high-magnification images. At a minimum, the authors should utilize stains that evaluate the integrity of the ECM, particularly in synovial compartments, ligaments and tendons of the foot (including Achilles tendon), bone, and cartilage. Stains such as safranin-O/fast green or toluidine blue for cartilage, picosirius red to evaluate collagen alignment by polarized light microscopy, and Masson's trichrome should be performed. Sections that include the Achilles tendon should also be shown in addition to the ones provided, and higher magnification images of the site of dislocation, especially the areas of aberrant ECM deposition seen adjacent to the calcaneal displacement, should be included. Low and high-res photos should be provided side-by-side for normal and affected mice.

The available microCT data provided is compelling but under-analyzed. In particular, could joint space narrowing be quantified and/or bone mineral density of affected bones in the foot? Also, how is the shape of the bones affected compared to mice without the deformity - could this be quantified in some way?

We were not able to analyze the microCT data further as it is fell outside the scope of this paper. 

The use of a frailty score comprising 27 individual observational scores is uninformative. The authors should present these 27 scores individually in a Table to accompany the combined Frailty Score given in Figure 4, along with the appropriate statistical tests for each. In addition, how they were collected, including room conditions, the gender of the observer, whether the observer was blinded, and whether multiple observers were used, should be included in the Methods section. This type of information should be added for all behavioural assays used.

Please see supplemental Table 2 for the frailty score information. Please note additional information was added in the methods section for providing further details on the behavioral tests. I would like to note that experimenters are validated within the team. 

Overall, the behavioural data is interesting, and the pain contribution is compelling based on the analysis of Morphine administration. However, have the authors evaluated whether the innervation of the foot is altered? Or is inflammation elevated in affected animals?

We were not able to evaluate either the inflammation or innervation in the scope of this paper, but the author thanks the reviewer for these additional thoughts.

---

## [Decision Letter · Decision Letter 1]

1 Jun 2023

Evaluation of Tarsal Injuries in C57BL/6J Male Mice

PONE-D-22-26026R1

Dear Dr. Brenda Kick,

We’re pleased to inform you that your manuscript has been judged scientifically suitable for publication and will be formally accepted for publication once it meets all outstanding technical requirements.

Kind regards,

Joohyun Lim

Guest Editor

PLOS ONE

Additional Editor Comments (optional):

Reviewers' comments:

Reviewer's Responses to Questions

**Comments to the Author**

1. If the authors have adequately addressed your comments raised in a previous round of review and you feel that this manuscript is now acceptable for publication, you may indicate that here to bypass the “Comments to the Author” section, enter your conflict of interest statement in the “Confidential to Editor” section, and submit your "Accept" recommendation.

Reviewer #2: All comments have been addressed

2. Is the manuscript technically sound, and do the data support the conclusions?

Reviewer #2: Yes

3. Has the statistical analysis been performed appropriately and rigorously? 

Reviewer #2: Yes

4. Have the authors made all data underlying the findings in their manuscript fully available?

Reviewer #2: Yes

5. Is the manuscript presented in an intelligible fashion and written in standard English?

Reviewer #2: Yes

6. Review Comments to the Author

Reviewer #2: The authors have addressed all major concerns. It would still be helpful to show the behavioural metrics separately as opposed to together in a cumulative score, but it is not necessary for acceptance at this time.

7. PLOS authors have the option to publish the peer review history of their article (what does this mean?). If published, this will include your full peer review and any attached files.

Reviewer #2: No

---

## [Editor Report · Acceptance letter]

15 Jun 2023

PONE-D-22-26026R1 

Evaluation of Tarsal Injuries in C57BL/6J Male Mice 

Dear Dr. Kick:

I'm pleased to inform you that your manuscript has been deemed suitable for publication in PLOS ONE. Congratulations! Your manuscript is now with our production department. 

Kind regards, 

on behalf of

Dr. Joohyun Lim 

Guest Editor

PLOS ONE